# Population Genomics, Virulence Traits, and Antimicrobial Resistance of *Streptococcus suis* Isolated in China

**DOI:** 10.3390/microorganisms13061197

**Published:** 2025-05-23

**Authors:** Yuying Li, Bin Ma, Xue Jia, Yanxi Wan, Shiting Ni, Guosheng Chen, Xin Zong, Hui Jin, Jinquan Li, Chen Tan

**Affiliations:** 1National Key Laboratory of Agricultural Microbiology, College of Veterinary Medicine, Huazhong Agricultural University, Wuhan 430070, China; liyuyingyy@webmail.hzau.edu.cn (Y.L.); mabin1996@webmail.hzau.edu.cn (B.M.); jinhui@mail.hzau.edu.cn (H.J.); 2Hubei Hongshan Laboratory, Wuhan 430070, China; 3The Cooperative Innovation Center for Sustainable Pig Production, Wuhan 430070, China; 4College of Biomedicine and Health, Huazhong Agricultural University, Wuhan 430070, China

**Keywords:** *Streptococcus suis*, virulence, antimicrobial resistance, genomics

## Abstract

*Streptococcus suis* is a significant zoonotic pathogen of public health importance. In this study, whole-genome sequencing of 177 isolates of *Streptococcus suis*, isolated from diseased swine across 15 provinces in China between 2017 and 2019, was performed. A total of 23 serotypes and 28 ST types were identified, with serotypes 2 and 3 comprising 50.8% of the isolates, and sequence types ST353 and ST117 accounting for 23.7%. Clustering analysis based on known virulence-associated factors (VAFs) resulted in the identification of four distinct clusters, and virulence was assessed using animal models, including a unique, highly virulent cluster designated as cluster I. Drug susceptibility testing indicated that 97.7% of the isolates were multidrug-resistant. A total of 26 resistance-associated genes were identified within the genome, 18 of which were associated with integrative and conjugative elements (ICEs) and/or integrative mobilizable elements (IMEs). Nevertheless, our understanding of *suis* virulence in terms of phylogeny remains incomplete. This study contributes to the understanding of the population structure and genetic characteristics of *suis*, provides a framework and novel partitioning approach for future investigations into its virulence and pathogenicity, and complements the data on antibiotic resistance.

## 1. Introduction

*Streptococcus suis* is a globally recognized zoonotic pathogen that significantly impacts the swine industry, leading to considerable economic losses. In swine, this bacterium is primarily responsible for conditions such as pneumonia, septicaemia, and meningitis [1]. In humans, *Streptococcus suis* infection is associated with disease primarily through exposure to contaminated byproducts or through occupational hazards [2,3]. China experienced major outbreaks of human infections in 1998, 2005, and 2016, all attributed to *Streptococcus suis* [4]. Between 2002 and 2013, over 1500 cases of human infection were documented across 34 countries [5]. Furthermore, *Streptococcus suis* serves as a significant reservoir for antimicrobial resistance (AMR) genes [6]. This bacterium exhibits a high prevalence of resistance to lincosamides [7], macrolides, and tetracyclines, with an increasing trend in resistance to sulfonamides, aminoglycosides, and fluoroquinolones over time [8]. Sequencing studies of *Streptococcus suis* have revealed the widespread presence of AMR-associated genes, many of which are located on integrative and conjugative elements (ICEs) and integrative mobilizable elements (IMEs) [9]. Additionally, *Streptococcus suis* has the capacity to transfer drug resistant genes to other pathogens, thereby exacerbating the issue of antimicrobial resistance [10].

Recent advancements in sequencing technology have led to a significant increase in the genetic data available for *Streptococcus suis*, primarily due to the whole-genome sequencing (WGS) of numerous isolates [11,12,13]. Multi Locus Sequence Typing (MLST) [14], which determines sequence types (STs) through the analysis of seven housekeeping genes, is widely employed to assess the genetic diversity and conduct epidemiological studies of *Streptococcus suis*. The most commonly isolated sequence types include ST1, ST25, ST28 [5], and ST7. Notably, ST1 is the predominant disease-associated sequence type in Europe, Asia, Africa, and South America [15], whereas ST25 and ST28 are more prevalent in North American isolates [16,17]. Conversely, ST7 is only endemic to mainland China [18,19]. In another study, a separate investigation analyzed SNPs from 1634 *Streptococcus suis* genomes utilizing Bayesian analysis of population structure (BAPS), resulting in the construction of a maximum-likelihood phylogenetic tree that delineates three primary clades: the human-associated clade (HAC), the diseased-pig clade (DPC), and the healthy-pig clade (HPC). The HAC includes several isolates that exhibit high virulence in experimental infection models [4].

Currently, 29 serotypes of *Streptococcus suis* have been identified on the basis of the antigenicity of capsular polysaccharide (CPS), specifically serotypes 1–19, 21, 23–25, 27–31, and 1/2 [20]. Recently, 28 new serotypes, referred to as NCL and Chz, have recently been recognized [21,22,23,24]. There are notable geographical variations in the distribution of these serotypes; however, serotype 2 predominates among isolates from various countries [5,25]. Serotypes 1, 3, 7, and 9 are the most frequently isolated serotypes from diseased pigs [26,27]. Serotypes and MLST types are often used to analyze the virulence phenotype of *suis*. Serotype 2 has been described as the most virulent serotype [5]. Conversely, it has been shown that isolates of serotype 2 prevalent in North America are less virulent [28]. In animal models of infection, ST1 was more virulent than ST25, and ST28 was the least virulent [16,29]. However, ST28 has reported different clinical presentationss in the United States [30]. Currently, more than 100 virulence-associated factors (VAFs) for *Streptococcus suis* have been proposed or validated, although the majority lack experimental validation [28,31]. Muramidase-released protein (MRP) [32], extracellular protein factor (EPF) [33], and suilysin (SLY) [34,35] are the most extensively studied VAFs believed to play a role in the pathogenesis of *S. suis*, and they are frequently utilized to predict the potential virulence of *S. suis* isolates. Nevertheless, the genotypes of *mrp*, *epf*, and/or *sly* do not definitively determine the virulence classification of *suis*, as strains lacking expression of these factors have still been found to exhibit virulent phenotypes [36]. In summary, *Streptococcus suis* appears to be a pathogen characterized by redundant virulence factors, making the identification of key virulence traits exceedingly challenging [28,37].

In this study, we performed genome sequencing on 177 isolates of *Streptococcus suis* obtained from diseased swine across 15 provinces in China during the period from 2017 to 2019. This analysis elucidated the population structure of *suis* and evaluated its potential virulence and antibiotic resistance. The 177 isolates were categorized into HAC, HPC, and DPC according to a previously reported scheme [4]. Furthermore, we examined these isolates in relation to 104 previously documented virulence-associated factors [31,38,39,40,41]. The virulence phenotypes of the distinct clusters within the three evolutionary clades were subsequently validated using a murine model. Overall, this study provides a framework for identifying potentially virulent strains among isolates and enhances the existing knowledge regarding the antibiotic resistance profiles of *Streptococcus suis* in China. Since the virulence phenotypes of *Streptococcus suis* cannot be fully characterized by a single or limited set of virulence factors [28], we wished to further understand the virulence of *suis* through phylogenetic analyses.

## 2. Materials and Methods

### 2.1. Streptococcus suis Isolate Collection

In this study, a total of 177 strains of *Streptococcus suis* were collected from various provinces across China between the years 2017 and 2019. Detailed information regarding the sources and characteristics of these strains can be found in the Appendix A. Efforts were made to obtain *Streptococcus suis* isolates from as many provinces as possible, ensuring that none of the isolates were duplicates derived from the same pig. The isolates were inoculated and cultured for 24 h on TSA supplemented with 10% bovine serum. Subsequently, monoclonal strains were selected and transferred to 5 mL of TSB medium, where they were incubated for 16 h at 37 °C in a 5% CO_2_ atmosphere.

### 2.2. Phylogenetic Analysis

To perform a wide-ranging phylogenetic analysis, 463 whole-genome sequences of *S. suis* were obtained from NCBI. In total, 640 *S. suis* sequences were collected for phylogenetic analysis. The sequences were first aligned to the reference genome (BM407) using MUMmer v. 3.23, and a total of 550,691 SNP sites were identified. In this study, indels and adjacent mismatches were not considered to be true SNPs. These SNPs of 640 *Streptococcus suis* sequences were subsequently used to construct the phylogenetic tree through IQ-TREE v. 2.2.0.3 with with the GTR + GAMMA model and 1000 bootstrap replicates. The clade information of each sequence was manually marked based on the phylogenetic tree and previous research. Finally, the phylogenetic tree was visualized by TVBOT and retouched using Adobe Illustrator (CC 2019).

### 2.3. MLST Analysis and Molecular Serotype Prediction

The sequence type of each isolate was annotated by SRST2 with default parameters. In addition, *S. suis* genome sequences were analyzed by BLASTn against a nucleotide database of serotype-specific genes in the CPS synthesis locus for molecular serotyping [42]. This identified 29 classical serotypes, with the exception of two pairs of serotypes: 1/14 and 2/$\frac{1}{2}$.

### 2.4. Identification of Virulence Factors (VFs) and Antimicrobial Resistance Genes (ARGs)

The VFs for each *S. suis* genome were determined using BLASTn against the database collected in this study by reviewing the references (Appendix A). The BLASTn minimum identity was set to 90%, and the minimum coverage was set to 80%. The ARGs of each strain were identified by RGI v. 6.0.3 using reference data from the Comprehensive Antibiotic Resistance Database (CARD). Here, Python 3.10.13 scripts were used to collect matching gene fragments for further analysis and visualization.

### 2.5. Annotation and Visualization of Bacterial Integrative and Conjugative Elements (ICEs)

The gene fragments of each strain identified as VFs or ARGs were selected for annotation with the ICEberg 3.0 database as ICE, ACE, IME, or CIME. In this study, BLASTn was employed to align each gene fragment with the ICEberg 3.0 database, setting the minimum identity at 90% and the minimum coverage at 80%. The resulting matches between gene fragments and ICE were then collected using Python scripts. Moreover, the datasets comprising VFs, ARGs, and ICEs from 177 *S. suis* genomes were visualized using the ggplot2 3.4.2 and ComplexHeatmap packages 2.15.4 in R 4.3.0, with subsequent editing conducted in Adobe Illustrator.

### 2.6. Antimicrobial Susceptibility Testing

The assessment of resistance was conducted by determining the minimum inhibitory concentration (MIC) values of *Streptococcus suis* isolates in relation to various antibiotic agents. The microbroth dilution method, as recommended by the Clinical & Laboratory Standards Institute (CLSI) in its 28th edition of M100, was employed for this purpose, and the results were interpreted in accordance with the established guidelines. A total of twelve antibiotics were utilized in this investigation, which were Ampicillin, Penicillin G, Cefepime, Florfenicol, Levofloxacin, Enrofloxacin, Minocycline, Clindamycin, Erythromycin, Clarithromycin, and Daptomycin.

### 2.7. Mouse Model of Infection

In this study, six-week-old female BALB/c mice were utilized and randomly assigned to various experimental groups. All procedures were sanctioned by the Ethics Review Committee for Animal Experimentation at Huazhong Agricultural University, adhering to stringent ethical guidelines to minimize animal suffering, which included the euthanasia of mice at the conclusion of the experiments. Bacterial cultures were prepared as previously described, inoculated in TSB medium supplemented with 10% bovine serum, and incubated at 37 °C in a 5% CO_2_ atmosphere until reaching the logarithmic growth phase. The bacterial suspensions were subsequently washed with phosphate-buffered saline (PBS), diluted appropriately in sterile PBS, and plated on tryptic soy agar (TSA) to quantify the viable bacteria after a 24 h incubation at 37 °C. For the mouse survival model, the mice were infected via intraperitoneal injection with 5 × 10^8^ colony-forming units (CFUs) and monitored daily for survival over a minimum period of five days. For the infection experiments, an intraperitoneal dose of 1 × 10^8^ CFU was administered as previously described. Following a 12 h infection period, the mice were euthanized through CO_2_ inhalation, and blood samples were collected via cardiac puncture. Lung tissues were harvested for the assessment of the bacterial load. Plasma was isolated from the blood through centrifugation at 4 °C for the analysis of cytokines, including TNF-α, IL-6, IL-12 p70, IFN-γ, and CXCL 1 (KC). Lung tissues were subsequently fixed in 4% paraformaldehyde, paraffin-embedded, and sectioned to 2–4 μm thickness for histopathological examination after hematoxylin and eosin staining.

### 2.8. Statistical Analysis

If not individually labelled, the data were analyzed using two-tailed, unpaired *t*-tests. The bacterial load, cytokine, and chemokine assays were all repeated at least three times, and the data are presented as the means. For all tests, a value of *p* < 0.05 was considered significant.

## 3. Results

This section may be divided by subheadings. It should provide a concise and precise description of the experimental results and their interpretations, as well as the experimental conclusions that can be drawn.

### 3.1. Serotype Distribution, MLST Identification, and Differential Clades

The 177 porcine isolates recovered in this study were obtained from clinical disease swine collected between 2017 and 2019, with each isolate originating from a different animal. The diseased tissues were obtained from 15 provinces in China, mainly from Hubei (*n* = 93/177) and Henan (*n* = 40/177) provinces (Appendix A). Isolates were recovered from the following anatomical sites: 70.0% (*n* = 124/177) from the lung; 18.0% (*n* = 32/177) from the brain; 4.5% (*n* = 8/177) from the joints; 1.7% (*n* = 3/177) each from the liver and effusion; 1.1% (*n* = 2/177) each from the spleen and kidney; and 0.5% (*n* = 1/177) each from the heart, intestine, and lymph nodes. Twenty-three serotypes were identified in the 177 isolates, with the majority of isolates being serotype 2 (*n* = 68, 38%), followed by those of serotype 3 (*n* = 22, 12%) and serotype 9 (*n* = 17, 12%). The remaining serotypes were distributed as serotypes 4 (*n* = 13), 7 (*n* = 11), 8 (*n* = 10), 1/2 (*n* = 9), and 19 (*n* = 4). The number of isolates in the remaining serotypes was less than or equal to 2 (Figure 1). According to the new typing scheme of the novel CPS loci (NCL) serotypes, one strain each of serotypes NCL17 and NCL3 on the new scheme was found. Both NCL17 and NCL3 were recovered from the lungs.

The top five isolates included ST353 (*n* = 25, 14.1%), ST117 (*n* = 17, 9.6%), ST1 (*n* = 17, 9.6%), ST28 (*n* = 16, 9.0%), and ST7 (*n* = 13, 7.3%), and the remaining STs were fewer in number (*n* < 10). In the following analysis, we identified 28 STs from the isolates, 22 of which were novel unique allele combinations (Figure 1). Serotype 2 consisted of six STs: ST1 (*n* = 16), ST7 (*n* = 13), ST25 (*n* = 4), ST28 (*n* = 7), ST242 (*n* = 4), and ST353 (*n* = 24). Serotype 3 consisted of four STs: ST27 (*n* = 2), ST108 (*n* = 2), ST117 (*n* = 17), and ST1004 (*n* = 1). Serotype 9 consisted of three STs and four novel STs; the known STs were ST243 (*n* = 7), ST1184 (*n* = 1), and ST1307 (*n* = 5). The phylogenetic tree (Figure 1) revealed that the serotype and MLST of *Streptococcus suis* isolates presented a high degree of diversity.

As previously defined by Dong et al. [4], large-scale genomic analysis on 1634 isolates of *Streptococcus suis* from a well-defined source has revealed three distinct bacterial clades, i.e., the healthy-pig clade, diseased-pig clade, and human-associated clade (HAC). There were still some isolates that had no clear clade (unclassified). Analyses using the reported protocols revealed in similar clades, i.e., HAC, DPC, and HPC (Figure 2).

### 3.2. Analysis of Virulence-Related Genes and Virulence Phenotypes

To further examine the relationship that could exist between virulence genes and virulence phenotypes of isolates, we characterized the carriage of VAFs in the genomes of *Streptococcus suis* isolates (*n* = 177). Among the virulence-associated factors of *Streptococcus suis*, *mrp*, *epf*, and *sly* are the most studied genes. The classical VAFs, *mrp*, *epf*, and *sly*, were identified in 72 (40.6%), 90 (50.8%), and 102 (57.6%) of the 177 isolates, respectively. We further examined the distribution of combinations of *mrp*, *epf*, and *sly* genes (Appendix A). These genes were dominated by *mrp*^+^*epf*^+^*sly*^+^ (60/177) and *mrp*^−^*epf*^−^*sly*^−^ (68/177), followed by *mrp*^−^*epf*^+^*sly*^+^ (28/177) and *mrp*^+^*epf*^−^*sly*^+^ (7/177), with no combined distribution of *mrp*^+^*epf*^+^*sly*^−^.

Given the limited distribution of classical VAFs (*mrp*, *epf*, and *sly*) among the isolates, we used 104 previously published possible VAF genes (including *mrp*, *epf*, and *sly*) in pigs and humans (Appendix A). Fifty-four VAFs were present in all 177 isolates, and the distribution of the presence of the remaining VAF genes is shown (Figure 3A). Clustering analysis was used to determine the relationship between the distribution of VAF and the isolates. The analysis of previously published possible VAF genes in pigs and humans identified four clusters (clusters I–IV) (Appendix A). Cluster II predominantly consisted of serotype 2 isolates, and 56% in cluster I were serotype 9. There was a high degree of overlap between cluster II and the HAC clade after clustering of the isolates, with all of cluster II (*n* = 60) occurring in the HAC clade. Cluster I (*n* = 25) appeared only in the HPC, and cluster III was distributed in the DPC (*n* = 46) and HPC (*n* = 4). Only one strain in cluster IV was from the HPC, and the remaining isolates (*n* = 41) were from the unknown clade (Appendix A). Interestingly, isolates with 100% mortality in HPC (Appendix A) all appeared in cluster I, while isolates with 40–60% mortality appeared in cluster III of HPC, and the isolates that did not cause mouse death were in cluster IV of HPC (*n* = 1).

To understand the virulence profile of the strains in the different clusters, we selected five isolates in cluster Ⅰ (Appendix A), four isolates assigned to cluster II, seven isolates assigned to cluster III, and four strains assigned to cluster Ⅳ, for a total of twenty isolates (Figure 3B). The survival of mice inoculated intraperitoneally with a 5 × 108 CFU dose (*n* = 5) of the different isolates until 7 days post-infection was assessed. We found that 18 isolates caused the death of the mice (Figure 3B), and that all infected mice presented clinical signs; the strains in the subgroups that did not cause death in mice were from cluster II and cluster Ⅳ. All of the mice died within 72 h of intraperitoneal injection of *Streptococcus suis* isolates (Appendix A), during which mortality was the result of sepsis and/or septic shock. *Streptococcus suis* isolates that caused 100% mortality were present in clusters Ⅰ, II, and III. Five isolates of cluster Ⅰ caused 100% mortality in mice, and there was no difference from SC19.

### 3.3. Mouse Model of Infection

The mortality caused by *Streptococcus suis* isolates may be the result of a combination of high blood/tissue bacterial titers and a systemic inflammatory response. Consequently, the bacterial burden in the blood, lungs, and brain, and the levels of cytokines in the serum were measured at 12 h post-infection. Significant differences were observed between cluster I-infected mice and cluster III-infected mice in lung tissue and blood (Figure 4B), and bacterial titers in cluster IV-infected mice were lower than those in the other two groups. The bacterial number of cluster I-infected mice was high, the number of cluster III-infected mice was intermediate, and the bacterial number of cluster IV-infected mice was low (Figure 4A). Infection of mice with cluster I strains (Z7590 and E7885) resulted in significant organ damage and signs of congestion and infiltration of inflammatory cells into the lungs (Figure 4C). One of the strains, strain E7885, was more clearly demonstrated in the lung tissue. The lung tissue damage and inflammatory cell infiltration caused by the cluster III isolates were not as severe as those caused by cluster I isolates, and no significant tissue damage or inflammatory cell infiltration was observed with the cluster IV isolates.

Excessive inflammation may also lead to death in mice. Cytokines and chemokines were measured in the plasma 12 h after the different strains were used to infect the mice (Figure 4D). The strains of cluster I all induced higher levels of interferon (IFN)-γ than those of cluster III (*p* < 0.001). Strain Z7590 in cluster I and both strains in cluster III showed significant differences in interleukin (IL)-6 and tumor necrosis factor (TNF)-α (*p* < 0.05), whereas strain E7885 did not. There were no significant differences in strains between the same VAF clusters. The chemokine C-X-C motif ligand (CXCL)1 was also detected. Interestingly, mice in cluster III with moderate mortality produced higher levels of CXCL1; specifically, strain E3111 produced 3 × higher CXCL1 expression than strain E7885 (*p* < 0.001). E3111 induced the lowest levels of IFN-γ, TNF-α, IL-6, and CXCL1.

### 3.4. Antibiotic Resistance Phenotypes and Genes

To assess the antibiotic resistance of the collected *Streptococcus suis* isolates, we conducted susceptibility tests against twelve antibiotics of seven different species of these isolates. All of the isolates were resistant to at least one antibiotic, and no isolates resistant to all twelve antibiotics were found at this time. Multi-resistant (resistant to antibiotics from at least three classes) strains accounted for 97.7% of the isolates, with 62.7% of the isolates (*n* = 111) being resistant to three classes of antibiotics (Figure 5A,B). A large proportion of these strains were resistant to Tetracycline (99.4%, *n* = 176), Clindamycin (97.1%, *n* = 172), Erythromycin (98.3%, *n* = 174), and Clarithromycin (97.1%, *n* = 172), and a small proportion of them were resistant to Ampicillin (9.6%, *n* = 17) and Cefepime (8.4%, *n* = 15). Further, we analyzed the resistance phenotypes of isolates classified as HAC, HPC, and DPC (Figure 5C). Isolates assigned to the HPC clade showed significant differences in drug resistance from the other clades. In the HPC clade, cluster III with intermediate mortality was more resistant than cluster I with high mortality, and they were significantly different from each other (*p* < 0.05). No significant differences in resistance phenotypes were observed among isolates from the HAC, DPC, and unknown clades.

To understand the genomic basis of the drug resistance phenotype of *Streptococcus suis*, we identified the presence of ARGs in the genomic sequence (Figure 6A). We found that the *Streptococcus suis* genome is abundant in antibiotic resistance genes, with a total of 26 ARGs identified (Appendix A), 18 of which were carried by ICE and/or IME (Figure 6B and Appendix A). The ARGs were identified as six aminoglycoside genes, (*APH(3*′*)-IIIa*, *SAT-4*, *aad(6)*, *ANT(6)-Ia*, *AAC(6*′*)-Ie-APH(2*″*)-Ia*, and *ANT(9)-Ia*), and seven tetracycline genes, (*tet(O)*, *tet(40)*, *tet(M)*, *tet(W)*, *tet(32)*, *tet(O/W/32/O)*, and *tet(L)*). All 177 strains carried the resistance gene for Quinolones was *patB*. The Gene *ErmB* for Macrolides accounted for 92.0% in all isolates, *tet(O)* for Tetracyclines accounted for 90.3%, the Gene *patA* for Quinolones accounted for 89.2% in all isolates, and other resistance genes were detected in less than 50% of cases. The most resistance genes carried by ICE and IME were *ErmB* and *tet(O)*. Moreover, 92.9% of the strains identified as carrying *tet(40)* were serotype 2, with *APH(3*′*)-IIIa* in 90%, *SAT-4* in 92.3%, and *aad(6)* in 97.1%.

## 4. Discussion

*Streptococcus suis* is a zoonotic pathogen with a worldwide presence [43]. *Streptococcus suis* is a highly diverse species [44], with the current findings identifying 23 serotypes and 28 sequence types among the isolates. Notably, serotypes 2 and 3 were the most frequently isolated from clinical cases in Canada [5], whereas another investigation indicated that serotypes 4 and 7 were more prevalent during the period of 2015–2016 compared to 1996–2004 [44]. ST7 was found only in China [19], with all ST7 strains classified as serotype 2. Additionally, four of the twenty-two strains representing novel sequence types were identified as serotype 9, whereas the remaining strains included two each of serotypes 16, 28, 29, and 30. Serotype 9 exhibited considerable genetic heterogeneity and diversity [45,46].

Previous studies have delineated three distinct clades of *Streptococcus suis*: the healthy-pig clade (HPC), the diseased-pig clade (DPC), and the human-associated clade (HAC) [4]. Analysis of isolates with known origins revealed that 96% of human isolates were classified within the HAC, 72.2% of isolates from healthy pigs clustered in the HPC, and 69% of isolates from diseased pigs were categorized in the DPC [4]. The isolates of *Streptococcus suis* obtained from sick pigs in this study were similarly distributed among the HAC (34.5%), HPC (16.9%), and DPC (26%) clades. Furthermore, genomic data from tonsils of clinically healthy pigs [12] revealed a distribution of 2.7% in HAC, 90.1% in HPC, and 2.2% in DPC (Appendix A). These findings suggest that isolates derived from diseased pigs may possess a richer genomic repertoire.

Muramidase-released protein (*mrp*), extracellular protein factor (*epf*), and suilysin (*sly*) represent the most extensively researched virulence-associated factors (VAFs) and have been utilized as indicators of the virulence profile of *Streptococcus suis* [28]. In clinical disease samples collected in Canada, isolates were found to be negative for *mrp* [11], and the absence of *mrp* was predominantly noted in serotype 9 isolates from affected pigs [47]. While *mrp* is correlated with virulent strains of ST1, the results are not the same for other serotype 2 STs [16]. Additionally, studies utilizing mouse intraperitoneal infection models indicated that bacterial burdens were significantly greater in *sly*-positive strains [48,49], and when employing an intranasal mouse model for mucosal colonization, the bacterial burden of an isogenic *sly* mutant did not differ significantly from that of the wild-type strain [50]. Findings from the porcine intranasal infection model suggest that *mrp* functions more as a “virulence marker” rather than a definitive “virulence factor” [51]. In the present study, despite the isolates being derived from diseased pigs, *mrp* was absent in 40.7% of the isolates, *epf* was absent in 49.1%, and *sly* was absent in 42.4%, with a combined *mrp*^−^*epf*^−^*sly*^−^ genotype present in 38.4% of cases. In contrast, numerous isolates from healthy pigs presented the *epf*^+^*mrp*^+^*sly*^+^ genotype [12]. Importantly, for many pathogens, the knockout of specific genes or proteins does not yield significant phenotypic changes, and in some instances, the functions may be preserved, thereby complicating the analysis of virulence factors due to redundancy [28]. In conclusion, *Streptococcus suis* is a pathogen characterized by virulence factor redundancy, and virulence-associated factors in conjunction with transcriptomic and proteomic methods can more effectively elucidate the regulatory mechanisms of virulence in future studies.

On the other hand, the virulence of *Streptococcus suis* exhibits notable heterogeneity, necessitating infection experiments to validate the virulence of various strains [28]. In this study, our isolates were categorized into clusters I–IV based on virulence-associated factor (VAF) cluster analysis. Cluster II completely overlaps with the HAC, with serotype 2—recognized as the most prevalent serotype responsible for human infections—comprising 86.4% of this cluster. Cluster II may be linked to human infections caused by *suis*, and it is characterized by the greatest number of identified virulence factors. While the genomic size of human-associated *suis* isolates is diminishing, there is a tendency for the encoding of known or putative virulence factors to occur at high frequencies in pathogenic isolates [4,52]. The virulence-related genes, *hylA*, *PnuC*, *rfeA*, *Sbp1*, and *Sbp2*, only appeared in cluster II, and these genes are considered as putative zoonotic virulence factors [39]. The HPC not only illustrates evolutionary diversity within the phylogenetic tree but also encompasses three additional clusters beyond cluster II in the VAFs. Cluster I, which is exclusive to the HPC, demonstrates high virulence and is associated with elevated mortality rates in animal models. The *1901HK/RR*, *endoSS*, and *GH 92* genes were present in cluster I but not detected in cluster III. The *1901HK/RR* is associated with adherence and invasion, aiding *suis* in surviving in the bloodstream [53]. *EndoSS*-inserted *suis* exhibit a higher proliferative capacity, while *GH 92* can degrade N-glycans to generate *endoSS* substrates [54]; these VAFs may facilitate higher loads of suis in host blood and tissues. Although the number of VAFs identified in cluster I was not the highest, this may be due to differences in the expression of genes highly correlated with virulence [38], or it is also possible that the lack of transcriptional regulators, combined with an increase in genes encoding DNA replication and repair proteins in the genome [55], resulting in a massive increase in bacterial populations within the host [52,55] and ultimately the death of the host. Compared with those from clusters III and IV, strains Z7590 and E7885 from cluster I presented significantly greater bacterial loads in the lungs and bloodstream and induced elevated levels of IL-6, TNF-α, and IFN-γ in the lungs and blood of infected mice, which, in turn, leads to the development of inflammatory diseases, such as fever and septic shock [56]. An excessive inflammatory response and high bacterial load may trigger sepsis [57], which is a leading cause of death [58]. Strains Z7590 and E7885 induced high levels of proinflammatory mediators. However, the medium mortality strain E3606 induced increased expression of CXCL1 chemokines. CXCL1 is known to activate neutrophils and enhance their phagocytic and bactericidal capabilities [59], and this may result in the premature clearance of E3606 by neutrophils before it can cause damage to the lungs or other tissues, thereby enhancing survival in mice. Significant tissue damage and cellular infiltration were observed in tissue sections following infection with strain E7885, thereby increasing the likelihood of mortality in mice [60]. Although the HPC represents an evolutionary clade associated with the colonization of healthy pigs, the strains classified within it retain the potential to cause disease and include highly virulent cluster I strains.

The findings from the Minimum Inhibitory Concentration (MIC) analysis revealed that a significant proportion of *Streptococcus suis* isolates are categorized as multi-drug resistant (MDR). Notably, strains within the HPC exhibit heightened resistance levels, and the presence of antibiotic pressure appears to facilitate conditions conducive to host colonization [61]. The HPC clade demonstrates increased resistance within cluster III, which is associated with moderate virulence, while strains exhibiting lower virulence may confer certain advantages [62]. Of particular interest is the strain E7885 from cluster I lineage, which is characterized by both high virulence and resistance to ten different antibiotics. *Streptococcus suis* is a repository of resistance genes [8], and more than half of the resistance genes in *Streptococcus suis* are carried by ICE and/or IME [6,9]. In this study, 73.1% of the resistance genes identified were located in ICE and or IME. Notably, despite the prohibition of chloramphenicol usage in animal production facilities in China since 2002 [63], the chloramphenicol resistance genes, *catA8*, *fexA*, and *cat-TC,* were identified. The data presented in this report underscore the genetic diversity present within *Streptococcus suis*. This study also assessed the virulence-associated factors (VAFs) and resistance genes among various isolates. The MIC analysis established the resistance profiles of the strains and corroborated the virulence of the isolates in animal models.

## Figures and Tables

**Figure 1 microorganisms-13-01197-f001:**
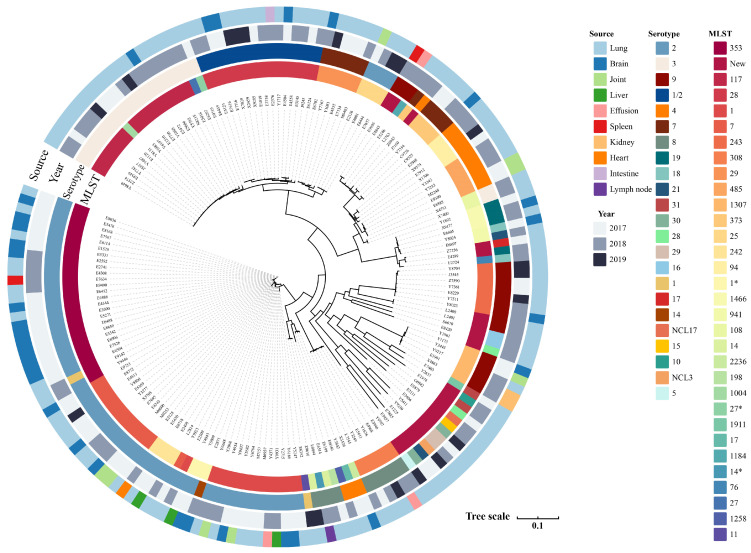
Phylogenetic tree of 177 *Streptococcus suis* isolates. This phylogenetic tree was generated utilizing genome-wide SNPs. The legends, arranged from the innermost to the outermost ring, include the following information: ST type, serotype, isolation time, and tissue source. Those containing at least one mismatched gene are marked with an asterisk.

**Figure 2 microorganisms-13-01197-f002:**
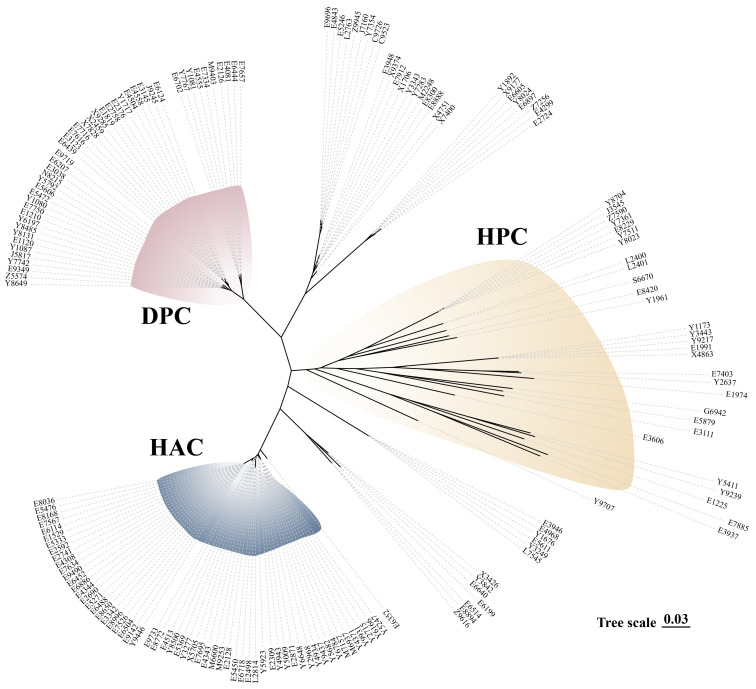
Phylogenetic tree (unrooted) showing the clades of 177 isolates of *Streptococcus suis* isolates, with HAC denoted in blue, HPC in yellow, DPC in red, and the remaining isolates classified as an unknown clade, indicated by the absence of color.

**Figure 3 microorganisms-13-01197-f003:**
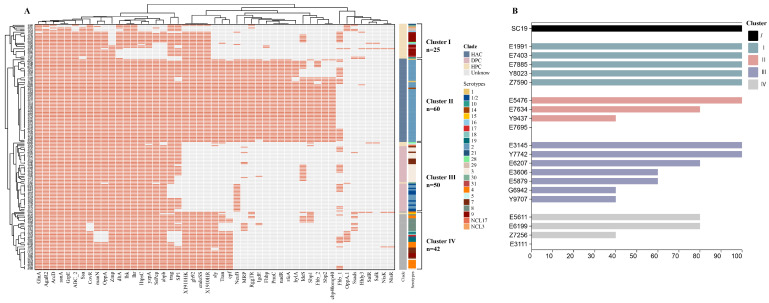
(**A**) Virulence associated factors (VAFs) in 177 isolates of *Streptococcus suis*. The heatmap illustrates the presence or absence of 104 previously identified virulence associated factors (VAFs) across 177 isolates of *Streptococcus suis*. The color coding indicates presence in orange and absence in grey. The right column delineates the clades and serotypes of the isolates, which are categorized into four distinct clusters (clusters Ⅰ–Ⅳ) based on their VAFs. (**B**) The mouse infection model was generated via intraperitoneal injection, involving a total of 20 isolates, which included 5 isolates from cluster Ⅰ, 4 from cluster II, 7 from cluster III, and 4 from cluster Ⅳ.

**Figure 4 microorganisms-13-01197-f004:**
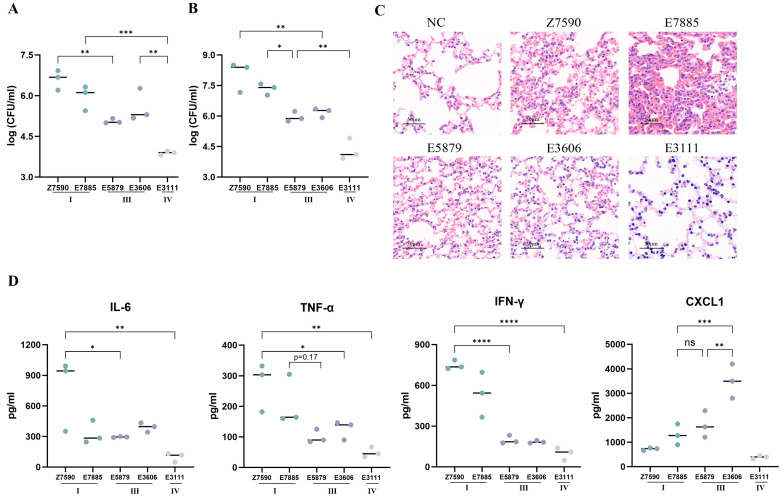
Bacterial load and lung tissue of mice infected with *suis* of cluster I, cluster III. and cluster IV in HPC. The mice were injected peritoneally with 5 × 10^8^ CFU for 12 h. Bacterial loads in (**A**) blood and (**B**) lungs. (**C**) Histological changes in the lungs. (**D**) Cytokines and chemokines in cluster I, cluster III, and cluster IV of HPC *Streptococcus suis*-infected mice. Mice (*n* = 3) were injected peritoneally with 5 × 10^8^ CFUs, and IL-6, TNF-α, IFN-γ, and CXCL1 levels were quantified via ELISA 12 h later. Each dot represents one mouse. In the figure, ‘*’ represents *p* < 0.05, ‘**’ represents *p* < 0.01, ‘***’ represents *p* < 0.001 and ‘* ***’ represents *p* < 0.0001.

**Figure 5 microorganisms-13-01197-f005:**
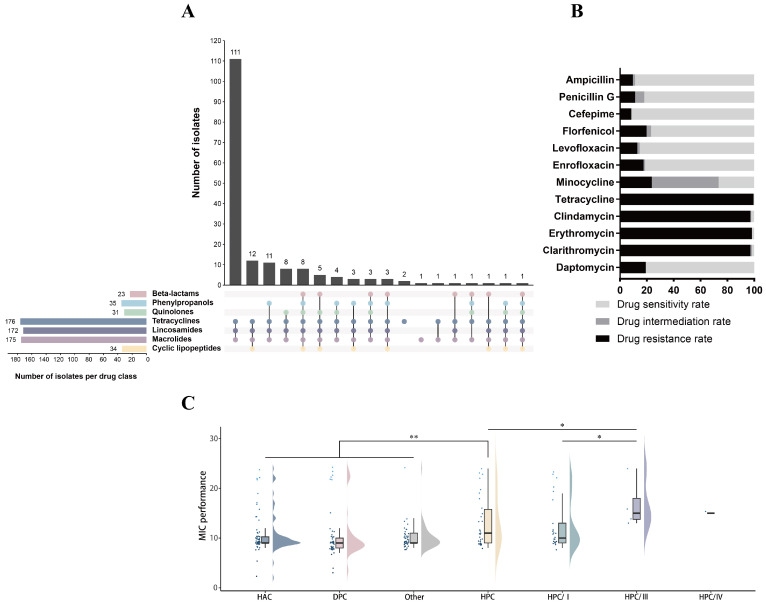
(**A**) Antibiotic resistance upset plot for *Streptococcus suis*, with horizontal bars showing antibiotic species and the number of isolates. The colored circles represent resistance to one antibiotic. Vertical bars represent the number of isolates with a particular combination of resistance. (**B**) Resistance ratios (black), mediator ratios (grey), and susceptibility ratios (light grey) are shown for each antibiotic drug. (**C**) Resistance phenotypes for each of the clades. Resistant = 2, mediator = 1, and sensitive = 0. The vertical axis is the sum of the values. A single asterisk (*) indicates *p* < 0.05 and double asterisks (**) indicates *p* < 0.01.

**Figure 6 microorganisms-13-01197-f006:**
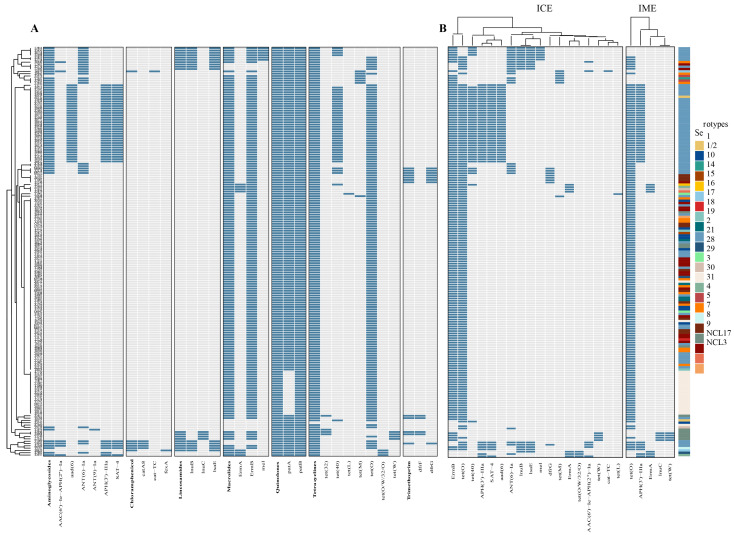
(**A**) Heatmap showing the presence (blue) or absence (grey) of resistance genes. The horizontal axis represents the drug resistance genes. (**B**) Resistance genes carried by ICE and IME. The columns on the right represent serotypes.

## Data Availability

The original contributions presented in the study are included in the article/Appendix A, and further inquiries can be directed to the corresponding authors.

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
