# Peer review of "Population Genomics, Virulence Traits, and Antimicrobial Resistance of Streptococcus suis Isolated in China"

_microorganisms, 2025, doi:10.3390/microorganisms13061197_

Round 1

Reviewer 1 Report

Comments and Suggestions for Authors

This study addresses an important zoonotic pathogen using robust genomic and phenotypic approaches. The findings on Streptococcus suis virulence clustering and antimicrobial resistance profiles are valuable contributions to the field. However, the Discussion section would benefit from further clarification and deeper critical analysis, particularly regarding:
 • the implications of the virulence–resistance relationship,
 • the interpretation of gene presence versus gene expression,
 • and more thorough inter-cluster comparisons.

While the study is presented logically, several overarching issues remain, in particular the overgeneralisation of virulence factors. Specific points are listed below:

  • Lines 395–396: The conclusion that virulence factor identification is “almost unattainable” overstates the challenge. While redundancy complicates analysis, transcriptomic and proteomic methods are revealing key regulatory mechanisms. Consider offering a future perspective instead of a definitive statement.
  • There is a limited comparison of clusters beyond Cluster I (Lines 397–427); In this section would be beneficial expanding the paragraph by:
    - Comparing gene content and potential regulatory mechanisms between clusters.
    - Clarifying if mortality in Cluster III (intermediate virulence) might be linked to modulation of immune signalling.
  • Unclear hypothesis development (Lines 411–417). There are valuable insights, but the section needs clearer phrasing and referencing.
  • Ambiguity in linking inflammation to survival (Lines 416–421)
  • Resistance-Virulence connection (Lines 428–435). E7885 is both highly virulent and multidrug-resistant. This raises key questions:
     - Is this combination rare or increasingly common?
     - What might explain the coexistence of fitness and resistance traits?
  • “Passive loss of regulatory function” (Line 411) is vague. Do you mean loss of function mutations in repressors, or derepression via ICE acquisition?
  • Avoid repeating “highly virulent” and “significantly” too often without quantitative comparison. Instead, integrate values or effect sizes (for example 5× higher IFN-γ expression).
  • “This study also assessed…” (Line 442) is redundant. The preceding sentence already summarises the contribution. Consider removing or tightening this part.
  • Consistently italicise Streptococcus suis (e.g., Line 36, Line 114).
  • Avoid the repetitive structure. - Revise to: “China experienced major outbreaks of human infections in 1998, 2005, and 2016, all attributed to Streptococcus suis. - Line 100: “understanding of suis virulence in the terms of phylogeny.”

  • Line 118: IQ-TREE usage is appropriate, but the substitution model used (e.g., GTR+G, HKY) should be indicated for transparency and reproducibility.

  • Line 40-41: check language repetition.

Reviewer 2 Report

Comments and Suggestions for Authors

Dear Authors, Thank you for uploading your manuscript titled "Population genomics, virulence traits and antimicrobial resistance of Streptococcus suis isolated in China." Due to the geographically restricted area of study, the topic is interesting for all researchers, even if more for Chinese researchers. The characterisation of virulence and AMR tracts is particularly interesting because of the widespread problem of AMR spreading.

Please control all over the entire text for the size of typing characters you use for writing the manuscript; the genes have to be put in italics because it is a worldwide rule.

 Moreover, the attached file shows the other comments. 
